# Estimating the global health impact of gender-based violence and violence against children: a systematic review and meta-analysis protocol

Cory N Spencer ,[1] María Jose Baeza,[2] Jaidev Kaur Chandan,[3] Alexandra Debure,[2] Molly Herbert,[1] Teresa Jewell,[4] Mariam Khalil,[1] Rachel Qian Hui Lim,[5] Sonica Minhas ,[5] Joht Singh Chandan ,[6] Emmanuela Gakidou ,[1] Nicholas Metheny [2]

JSC, EG and NM are joint senior authors.

For numbered affiliations see end of article.

**Correspondence to**
Cory N Spencer;
corycns@uw.edu

## ABSTRACT

**Introduction** Exposure to gender-based violence (GBV) and violence against children (VAC) can result in substantial morbidity and mortality. Previous reviews of health outcomes associated with GBV and VAC have focused on limited definitions of exposure to violence (ie, intimate partner violence) and often investigate associations only with predefined health outcomes. In this protocol, we describe a systematic review and meta-analysis for a comprehensive assessment of the impact of violence exposure on health outcomes and health-related risk factors across the life-course.

**Methods and analysis** Electronic databases (PubMed, Embase, CINAHL, PsycINFO, Global Index Medicus, Cochrane and Web of Science Core Collection) will be searched from 1 January 1970 to 30 September 2021 and searches updated to the current date prior to final preparation of results. Reviewers will first screen titles and abstracts, and eligible articles will then be full-text screened and accepted should they meet all inclusion criteria. Data will be extracted using a standardised form with fields to capture study characteristics and estimates of association between violence exposure and health outcomes. Individual study quality will be assessed via six risk of bias criteria. For exposure–outcome pairs with sufficient data, evidence will be synthesised via a meta-regression—Bayesian, regularised, trimmed model and confidence in the cumulative evidence assessed via the burden of proof risk function. Where possible, variations in associations by subgroup, that is, age, sex or gender, will be explored.

**Ethics and dissemination** Formal ethical approval is not required. Findings from this review will be used to inform improved estimation of GBV and VAC within the Global Burden of Disease Study. The review has been undertaken in conjunction with the Lancet Commission on GBV and the Maltreatment of Young People with the aim of providing new data insights for a report on the global response to violence.

**PROSPERO registration number** CRD42022299831.

## STRENGTHS AND LIMITATIONS OF THIS STUDY

⇒ This review is the first effort to systematically identify and assess all health-related impacts of multiple and overlapping forms of gender-based violence and violence against children.

⇒ Data analysis plans include meta-regression and burden of proof frameworks to synthesise all available evidence and provide policy-direct interpretations of associations.

⇒ Findings from the review will be incorporated in the Global Burden of Disease Study, the most comprehensive observational epidemiological study to date and a critical tool for researchers, advocates and decision-makers.

⇒ Challenges remain in the comparability of definitions of exposure to gender-based violence and violence against children and availability of high quality data for under-studied forms of violence (ie, cyberviolence, stalking, elder abuse).

## BACKGROUND

Gender-based violence (GBV) (including but not limited to intimate partner violence (IPV), elder abuse and violence against women (VAW)) and violence against children (VAC) are global public health issues associated with a substantial burden of morbidity and mortality. It is well known that the immediate consequences of both VAC and GBV in adulthood include physical injuries and death.[1] However, the medium-term and longer-term consequences are less well understood, but have shown to span a variety of physical, mental, sexual and reproductive health issues.[2 3] Until recently, the fields of VAC and GBV were largely siloed, stunting our understanding of how different exposures to violence influence each other across the lifespan. To address these challenges, the Sexual Violence Research Initiative, UNICEF Innocenti and the World Health Organization (WHO) have recently developed a framework of guiding principles encouraging

interaction of research in the field of violence epidemiology.[4] As international advocacy and research organisations push for the integration of these fields, a more fulsome understanding of the health impacts of violence across the life-course is needed.[5 6]

The Global Burden of Diseases (GBD), Injuries and Risk Factors Study has quantified the global disease and disability burden of two violence-related risk factors, IPV and childhood sexual abuse (CSA), within a comparative risk assessment framework since 2010.[7–10] An advantage of the comparative risk assessment framework is the ability to compare the relative contribution to disease and disability among several health risk factors. Indeed, country-specific and age-specific findings from the GBD have shown IPV to account for more overall disability-adjusted life-years (DALYs) in women ages 15–49 than more traditionally highlighted health risk factors such as smoking.[11 12] On a global scale, IPV was estimated to account for 6.44 million (95% uncertainty interval (UI), 3.55–9.87 million) DALYs among this group in 2019 while, by comparison, smoking contributed to 4.52 million (95% UI 3.87–5.23 million) DALYs in the same population in 2019.[8] Much of the estimated health impact stems from GBD meta-analyses of scientific literature, which have found IPV exposure to be associated with a 54% increased risk of depression and 60% increased risk for HIV infection.[8] Likewise, those exposed to CSA have been estimated to be 2.21 times as likely to experience alcohol use disorder (relative risk (RR)=2.21, 95% UI=1.15–4.04) and 1.56 times as likely to experience depression (RR=1.56, 95% UI=1.30–1.86), accounting for 3.67 million (95% UI, 1.75–6.56 million) global DALYs among males and females of all ages in 2019.[8] While these findings provide a basis for understanding the impact of violence on health, the lack of a comprehensive analysis of the longitudinal literature has so far precluded the ability to expand the types of violence included in the GBD as well as the specific health outcomes that comprise the final estimates of burden. A more complete understanding of the adverse health outcomes associated with exposure to more types of GBV and VAC, and the magnitude of these associations, is needed to capture the negative health and societal impacts of GBV and VAC.

Beyond the estimates provided by the internationally comparative GBD studies, existing reviews assessing the health impacts of GBV and VAC have typically focused on the impact of a single type of violence (eg, IPV) on a specific health outcome (eg, HIV). In 2013, the WHO, London School of Hygiene and Tropical Medicine, and South African Medical Research Council conducted a systematic review and meta-analysis of a variety of health effects related to specific forms of GBV, measured as physical and/or sexual IPV or non-partner sexual violence (NPSV).[2] Across studies identified, women exposed to IPV were 1.5 times as likely to become infected with HIV/AIDS and 1.97 times as likely to experience depression, among other adverse health outcomes.[2] A lack of comparable studies prevented meta-analysis for NPSV.[2] Following

the publication of 2013 report, the WHO curated an extensive database of studies describing the literature explaining the relationship between VAW and VAC with subsequent health outcomes.[13] However, the database has not updated summaries of high-quality evidence on the health impacts of GBV (VAW/IPV) since 2013, which is urgently needed to inform global health policy. In 2018, Bacchus *et al* additionally reviewed cohort studies that reported on all health outcomes and behaviours related to recent physical and sexual IPV exposure, finding evidence of a positive, bidirectional relationship between these types of IPV and depressive symptoms.[14] Yet, there are fewer reviews investigating exposure types beyond physical and sexual IPV (eg, psychological violence, coercive control, financial abuse, stalking), and those available often define even finer scopes by investigating relationships between narrowly defined forms of violence and narrowly defined health outcomes, for example, mental health and gynaecological morbidity.[15 16]

Similar to GBV, there have been attempts to synthesise the literature exploring the breadth of consequences following exposure to VAC and highlighting the relationship between exposure to childhood maltreatment (including CSA) and a wide variety of psychosocial and health outcomes.[17 18] Research into the consequences of VAC has more recently overlapped with the burgeoning literature base describing health outcomes secondary to adverse childhood experiences (ACEs), two of which include direct exposure to violence and the witnessing of parental IPV.[19] A recent comprehensive review highlighted the pervasive harms that ACEs place on health throughout the life-course[20] and in a secondary analysis found that within Europe and North America, a 10% reduction in ACE prevalence could equate to annual savings of 3 million DALYs or US$105 billion.[21] However, despite these efforts to capture the literature on VAC through either exploring childhood maltreatment, CSA or ACEs as the marker of exposure, included studies are often limited to exposure in high-income countries and exclude other forms of violence such as female genital mutilation, trafficking, forced marriage and cyberviolence.

While the existing literature has illuminated the significant health impacts of violence, critical evidence gaps remain. These include the quantification of the health burden of less-studied forms of violence, the health burden of violence in in lower-income and middle-income settings, as well as the integration of violence in childhood and adulthood as an intergenerational issue that could be more effectively measured using a life-course approach. The life-course approach as outlined in the Minsk Declaration essentially recognises that all stages of a person's life are intricately intertwined with each other, with the lives of other people in society, and with past and future generations of their families.[22 23] In order to do adopt this approach effectively when considering the health effects of GBV/VAC, we must consider that violence can occur at any stage in one's life (preconception to death) but also

that the impact of such event can be inter-generational and societal. Additionally, as highlighted through the reviews cited above, the current research trajectory often creates distinctions between GBV/VAC and other forms of life-course violence such as elder-abuse.[24] However, considering that GBV and VAC share risk factors, co-occur and can lead to compounding consequences across the life-course, there is a clear need to examine these phenomena in unison.

We present a systematic review and meta-analysis protocol to generate estimates of a comprehensive range of health impacts associated with exposure to GBV and VAC. Findings will contribute to the assessment of risk-outcome relationships and attributable burdens of disease within the GBD. To our knowledge, there has been no other systematic review and meta-analysis conducted with such a life-course approach across multiple types of violence.

## METHODS AND ANALYSIS
The presentation of our review design follows the Preferred Reporting Items for Systematic Reviews and Meta-Analyses Protocols (PRISMA-P) guidelines (online supplemental material 1).[25]

### Aims of the review
The aim of this review is to identify and synthesise all available data on the health impacts of exposure to any form of GBV and VAC. This data can then be used to assess risk–outcome relationships and quantify their contribution to global disease and disability burdens.

### Specific review questions
1. How does exposure to GBV and/or VAC impact health across the life-course?
2. What is the strength of evidence on the associations between exposure to GBV and VAC and different health impacts?
3. Do estimates of association vary by characteristics of the violence, global region, characteristics of the perpetrator and/or characteristics of the victim?

### Definitions
#### Definitions of violence
We include in our searches the following terms describing exposure to GBV and/or VAC:
- GBV.
- IPV, partner abuse/violence, wife/spouse abuse, dating abuse/violence.
- Sexual abuse, rape, forced sex, sexual assault, sexual coercion, sexual exploitation.
- Reproductive coercion.
- Female genital mutilation, female genital cutting, female circumcision.
- Sex trafficking, child, early and forced marriage.
- Physical abuse.
- Psychological abuse, emotional abuse, verbal abuse.
- Economic abuse, financial abuse.
- Cyberviolence, cybervictimisation.
- Domestic violence/abuse.
- ACEs that include direct exposure to and witnessing of violence.
- Child maltreatment, molestation, child abuse.
- Elder abuse, senior abuse, aged abuse.
- Stalking, cyberstalking.
- Dehumanisation, torture.
- Workplace violence, student abuse, sexual harassment.
- GBV perpetrated with a firearm.

We expect author definitions and methods used to measure exposure to vary and will accept all definitions, documenting study definitions and measurement techniques as a part of study-level quality assessment.

### Health outcomes
We did not restrict searches to predefined health outcomes and aim to accept all literature reporting an association between violence exposure and health. Definitions of health outcomes and health-related risk factors will be guided by cause, injury and risk factor case definitions from the GBD study.[8 26] Studies that report on certain biomarkers without accompanying clinical diagnoses (ie, neural connectivity patterns, salivary cortisol as a stress response, DNA characteristics) will not be eligible for inclusion. Similarly, studies that report on the presence or number of disease symptoms without an accompanying diagnosis of a health outcome will not be eligible for inclusion. Reviewers will meet regularly to raise questions about eligible health outcomes, with consensus decisions documented and circulated via written guidelines. Differences in measurement methods or case definitions of eligible health outcomes will be documented as a part of quality assessment as well. Final selection of associations to be synthesised will depend on the availability of studies that examine the association between a comparable form of exposure and reported health outcome.

### Criteria for considering studies for this review
#### Inclusion
- Study design: case–control, cohort or case-crossover studies.
- Participants: Studies conducted in participant groups likely to be generalisable to the population of interest. Exposed groups are defined as any individual who has experienced a form of GBV and/or VAC throughout the lifetime. Comparators will be non-exposed control groups, or study groups without reported exposure to a form of GBV and/or VAC.
- Outcomes: Studies reporting an estimate of association (either RR, risk ratio, odds ratio, hazard ratio or similar) or reporting cases and non-cases among those exposed and unexposed. If not provided directly, studies providing enough information to allow an estimate of RR to be calculated will meet inclusion criteria.

### Exclusion

► Study design: Cross-sectional, ecological, case series or case studies. We exclude cross-sectional studies in accordance with GBD study risk factor analyses, which typically do not include cross-sectional studies. This exclusion reason is related to the inability to assess temporality between exposures and outcomes in cross-sectional studies. We do not anticipate there to be any experimental studies, however, these will also be excluded.

► Participants: Studies conducted in subgroups identified only by convenience sampling or subgroups identified via a shared characteristic that is likely related to risk of exposure to violence or the reported health outcome (eg, domestic violence shelter residents).

► Exposure measurement: Studies that report only an aggregate measure of exposure combining exposure to a form of violence with other, non-eligible exposures (eg, reports a composite ACE score only) will be excluded. For these studies, we are unable to disentangle the effect of violence exposure from the effects of other hardships or exposure types, preventing their inclusion in our review.

► Does not meet minimum reporting criteria: Studies missing essential data, that is, those that do not report effect sizes and uncertainty information (confidence intervals, sample sizes) or the data needed to impute an effect size with uncertainty information.

► Studies reporting on the same exposure and outcome using the same data: The study with the longest follow-up time period or most complete dataset will be included.

### Search strategy for identifying relevant studies

PubMed, Embase, CINAHL, PsycINFO, Global Index Medicus, Cochrane and Web of Science Core Collection will be searched using controlled vocabulary and keyword search terms. All relevant studies published between 1 January 1970 and 30 September 2021 will be considered, regardless of language of publication or study setting. Immediately prior to preparing final results from the review and meta-analysis, searches will be updated to the current month to retrieve for inclusion any further studies identified. The search strategy for PubMed is provided in table 1. The search terms for Embase, CINAHL, PsycINFO, Global Index Medicus, Cochrane and Web of Science are provided in online supplemental tables 1–6, respectively (online supplemental material 2).

### Data management and extraction

Search results will be merged and duplicates removed using the systematic review management software Covidence.[27] All reviewers will initially screen the first 50 search results and meet to compare screening decisions and clarify any questions with regard to the inclusion criteria. The first two-thirds of titles and abstracts will be screened by two independent reviewers, and JSC, NM and CNS will review and resolve all conflicts that arise during

**Table 1** Search terms and strategy for PubMed

| Search terms | Concept |
|---|---|
| 1. "Sex Offenses"(mh). | Violence exposure |
| 2. "Violence"(mh:noexp). | |
| 3. "Domestic Violence"(mh). | |
| 4. "Gender-Based Violence"(mh). | |
| 5. "Intimate Partner Violence"(mh). | |
| 6. "Physical Abuse"(mh). | |
| 7. "Rape"(mh). | |
| 8. "Torture"(mh). | |
| 9. "Workplace Violence"(mh). | |
| 10. "Gun violence"(mh). | |
| 11. "Battered Women"(mh). | |
| 12. "Adult Survivors of Child abuse"(mh). | |
| 13. "Exposure to Violence"(mh). | |
| 14. "Emotional Abuse"(mh). | |
| 15. "Sexual Harassment"(mh). | |
| 16. "Harassment, Non-Sexual"(mh:noexp). | |
| 17. "Emotional abuse"(mh). | |
| 18. "Aggression"(mh:noexp). | |
| 19. "Dehumanization"(mh). | |
| 20. "stalking"(mh). | |
| 21. "adverse childhood experiences"(mh). | |
| 22. violence(tiab). | |
| 23. "sexual assault"(tiab). | |
| 24. "sexual harassment"(tiab). | |
| 25. "sexual abuse"(tiab). | |
| 26. "sex abuse"(tiab). | |
| 27. rape(tiab). | |
| 28. "forced sex"(tiab). | |
| 29. "sexual coercion"(tiab). | |
| 30. "reproductive coercion"(tiab). | |
| 31. "sex trafficking"(tiab). | |
| 32. "sexual exploitation"(tiab). | |
| 33. "forced marriage"(tiab). | |
| 34. "child marriage*"(tiab). | |
| 35. "early marriage*"(tiab). | |
| 36. "child bride*"(tiab). | |
| 37. CEFM(tiab). | |
| 38. infibulation*(tiab). | |
| 39. clitoridectom*(tiab). | |
| 40. clitorectom*(tiab). | |
| 41. "ritual female genital surger*"(tiab). | |
| 42. FGM(tiab). | |
| 43. "female genital mutilation"(tiab). | |
| 44. "female genital cutting"(tiab). | |
| 45. "female circumcision"(tiab). | |
| 46. "female genital circumcision"(tiab). | |
| 47. "physical abuse"(tiab). | |

Continued

**Table 1** Continued

| Search terms | Concept |
|---|---|
| 48. "psychological abuse"(tiab). | |
| 49. "emotional abuse"(tiab). | |
| 50. "economic abuse"(tiab). | |
| 51. "financial abuse"(tiab). | |
| 52. "verbal abuse"(tiab). | |
| 53. maltreatment(tiab). | |
| 54. "violent discipline"(tiab). | |
| 55. "corporal punishment"(tiab). | |
| 56. "adverse childhood experience*"(tiab). | |
| 57. molestation(tiab). | |
| 58. "child abuse"(tiab). | |
| 59. "partner abuse"(tiab). | |
| 60. "dating abuse"(tiab). | |
| 61. "wife abuse"(tiab). | |
| 62. "spouse abuse"(tiab). | |
| 63. "domestic abuse"(tiab). | |
| 64. "elder abuse"(tiab). | |
| 65. "senior abuse"(tiab). | |
| 66. "aged abuse"(tiab). | |
| 67. victimization(tiab). | |
| 68. dehumanization(tiab). | |
| 69. victimisation(tiab). | |
| 70. dehumanisation(tiab). | |
| 71. stalking(tiab). | |
| 72. cyberviolence(tiab). | |
| 73. cybervictimization(tiab). | |
| 74. cyberstalking(tiab). | |
| 75. Or/1–74 | |
| 76. Case-Control Studies(mh). | Study type |
| 77. Cross-Over Studies(mh). | |
| 78. Cohort Studies(mh). | |
| 79. Systematic Review(pt). | |
| 80. Meta-Analysis(pt). | |
| 81. "Twin Study"(pt). | |
| 82. "systematic review"(tiab). | |
| 83. "meta-analysis"(tiab). | |
| 84. "cohort"(tiab). | |
| 85. "cross-over"(tiab). | |
| 86. "case-control"(tiab). | |
| 87. "prospective"(tiab). | |
| 88. "retrospective"(tiab). | |
| 89. "longitudinal"(tiab). | |
| 90. "follow-up"(tiab). | |
| 91. "followup"(tiab). | |
| 92. Or/76–91 | |

Continued

**Table 1** Continued

| Search terms | Concept |
|---|---|
| 93. "Statistics as Topic"(mh). | Risk |
| 94. Risk(mh). | |
| 95. Odds Ratio(mh). | |
| 96. "risk*"(tiab). | |
| 97. "odds"(tiab). | |
| 98. "cross-product ratio*"(tiab). | |
| 99. "hazards ratio*"(tiab). | |
| 100. "hazard ratio*"(tiab). | |
| 101. statistic*(tiab). | |
| 102. "HR"(tiab). | |
| 103. "RR"(tiab). | |
| 104. "aOR"(tiab). | |
| 105. relation*(tiab). | |
| 106. correlat*(tiab). | |
| 107. associat*(tiab). | |
| 108. likel*(tiab). | |
| 109. Or/93–108 | |
| 110. "1970/01/01"(PDat). : "2021/09/30"(PDat). | Date restriction— all available literature since 1970 |
| 75 AND 92 AND 109 AND 110 | |

screening decisions. Upon confirmation of a low rate of conflicts (less than 10%) in the first two-thirds of double-screened articles, the remaining third of articles will be screened by a single reviewer. This approach balances the priorities of independent review and completing our review in a timely manner. Non-English publications will be reviewed using the language fluencies (Spanish, French and Portuguese) of the reviewers. Should articles in other languages be retrieved and eligible for extraction, the reviewers will contact colleagues fluent in these languages for assistance.

Reviewers will complete title and abstract screening of all articles before the team proceeds to full-text screening. Studies that met inclusion criteria in title and abstract screening will additionally be full-text screened and excluded if found to meet any of the exclusion criteria. Following the PRISMA guideline, each exclusion will include documentation of a specific exclusion reason. Within Covidence, there are several built-in exclusion reasons (ie, wrong study design, wrong setting, etc) as well as the possibility to create custom exclusion reasons. Reviewers will meet to discuss the addition of custom exclusion reasons prior to beginning full-text review, and will iteratively meet to discuss addition of new reasons as necessary. For full text review, 10% of articles will be reviewed by two independent reviewers and a meeting held to resolve misunderstandings and ensure all reviewers clearly understand inclusion and exclusion criteria. The remaining 90% of articles will be reviewed

**Table 2** Variables to be collected in the data extraction process

| Category | Data items |
|---|---|
| 1. Study characteristics | ► Author(s)<br>► Citation<br>► Year of publication<br>► Geographical location<br>► Study year(s)<br>► Study design<br>► Study name |
| 2. Population characteristics | ► Selection criteria<br>► Age summary statistics (range or median and standard deviation)<br>► Sex or gender summary statistics (percent women or percent female)<br>► Follow-up duration |
| 3. Exposure and outcome measurement | ► Exposure definition<br>► Exposure assessment frequency<br>► Exposure recall period<br>► Exposure type included in relative risk estimation<br>► Exposure categories<br>► Outcome definition |
| 4. Effect size and uncertainty | ► Effect size measure (eg, relative risk, odds ratio, incidence rate ratio, hazard ratio)<br>► Effect size<br>► Confidence interval and level<br>► Non-confidence interval uncertainty type and value<br>► Sample size (total, exposed, unexposed)<br>► Person-time (total, exposed, unexposed)<br>► No of events (total, among exposed, among unexposed)<br>► Whether main or subgroup analysis<br>► Description of subgroup analysis |
| 5. Risk of bias (quality assessment) | ► Exposure assessment method<br>► Exposure assessment instrument<br>► Outcome assessment method<br>► Outcome assessment instrument<br>► Representativeness<br>► Confounders adjusted for in reported effect size<br>► Percent of participants for which exposure data was obtained (for case-control studies)<br>► Drop-out rate (for cohort studies)<br>► Risk of reverse causation |

by one independent reviewer. If reviewers are unable to access the full text of a publication, the reviewers will reach out directly to the corresponding author and wait a maximum of 1 month for response, after which point the article will be excluded.

Data extraction will occur in parallel with full-text review, with some team members beginning to extract data once a sufficient number of full-text articles have been accepted. Before any reviewer begins data extraction, the entire review team will conduct a consensus building exercise in which all reviewers extract data from the same 10 accepted articles. In a group meeting, extractions will be compared and any questions resolved. Reviewers will extract data from accepted articles using a standardised form created in Covidence.[27] The data extraction form will include variables related to (1) characteristics of the study, (2) characteristics of the study population, (3) exposure and outcome measurement, (4) effect size and uncertainty, (5) risk of bias (quality assessment).[28 29] Data items are provided in table 2.

If a study reports on multiple forms of violence exposure, multiple associated health outcomes or reports findings by subgroup or model specification, data pertaining to each subanalysis will be extracted in addition to any aggregate results. In the case of a study reporting effect sizes for multiple model specifications, the most appropriately adjusted model will be selected for inclusion in meta-analysis.

### Risk of bias in individual studies
Sources of bias will be assessed and collected during data extraction. Following the Grading of Recommendations, Assessment, Development and Evaluations approach,[29] risk of bias criteria for individual studies include:
► Exposure measurement: How exposure to violence was assessed (whether standard, acts-based and specific questions were asked, eg, 'Have you ever been shoved, slapped, hit, or kicked by an intimate partner?' versus questions that rely on participants' own definition of abuse, eg, 'Have you ever experienced domestic

abuse?'). In addition, whether exposure was based on self-reports or another source (eg, health records).

► Outcome measurement: How reported health outcome(s) were measured (by physician diagnosis, diagnostic survey instruments, or electronic health records).

► Representativeness of study population: If a study sample was based on the general population or if study results are reported from a sub-group for which there are prior reasons to believe that findings would be different.

► Control for confounding: If a study statistically controlled for confounding using all major known confounders, including age, sex, education, income and other critical determinants of the health outcome.

► Selection bias: If a study is at risk of selection bias, based on per cent follow-up for longitudinal study designs and based on the percentages of cases and controls for which exposure data can be ascertained for case–control designs.

► Reverse causation: If a study is at risk of reverse causation, evaluated through study design and opportunity for recall bias (ie, case–control studies).[30]

### Data synthesis

If there are at least three studies identified with a comparable form of exposure and reported health outcome, we will synthesise effect sizes using a meta-regression—Bayesian, regularised, trimmed (MR-BRT) model.[8 31]

For each risk–outcome pair identified, we will use the MR-BRT tool to perform a meta-regression analysis of the risk of the given outcome for those exposed to the violence type relative to the reference category of those not exposed to the violence type. For risk–outcome pairs with sufficient data points, we will introduce likelihood-based trimming to detect and remove outliers before fitting the model, with an inlier fraction of 90%.[31]

For each risk–outcome pair meta-analysis, we will consider study-level covariates with the potential to bias the study's reported effect size estimates and adjust for these covariates if they are found to significantly bias the estimated RR. The MR-BRT tool includes an automated covariate selection process using a Lasso strategy to identify statistically significant covariates at a significance threshold of 0.05.[31 32]

The MR-BRT tool quantifies between-study heterogeneity by accounting for heterogeneity uncertainty and small numbers of studies.[31] In this approach, the Fisher information matrix is used to estimate uncertainty of the between-study heterogeneity parameter, $\gamma$.[31 33] The final uncertainty estimate reflects both the posterior uncertainty corresponding to the fixed effect and the 95% quantile of $\gamma$, which is sensitive to the number of studies, study design and reported uncertainty of the effect size.[31]

For each risk-outcome pair, we will additionally test for and report publication bias in the input data based on the Egger's Regression strategy, which tests the degree to which standard error is correlated with effect size in the data, and present funnel plots.[34 35]

### Additional analyses

If meta-analysis is not possible with all studies, we will synthesise the included study findings graphically[36] following the systematic review without meta-analysis guidance. This will include forest plots, which will graphically depict all study effect estimates using a single metric (eg, percent change) for each available health outcome and type of violence.[37] To produce the forest plots, we will transform effect estimates to a comparable metric wherever possible (ie, where the necessary data are available in the paper or from the authors). Harvest plots will demonstrate where inequities based on, for example, age of exposure, low-income and middle-income country versus high-income country, gender identity, ethnicity/race, sexual orientation, urban–rural location, exist in the available data.[38] When the necessary data are missing, all study effect estimates will be summarised in supplementary tables and discussed as relevant in text. If the necessary data are available (standardised effect estimate, p value) we will also consider albatross plots to summarise results.[39]

### Confidence in cumulative evidence

Confidence in risk–outcome pair results will be assessed via the burden of proof risk function (BPRF) methdology developed by GBD 2020 Risk Score Collaborators (unpublished methods). For a harmful risk, the BPRF is the 5% quantile risk function interpreted as the lowest level of risk consistent with available evidence. The average BPRF values across exposure observed in the studies will be summarised into star-rating categories, which are a policy-direct way to interpret the evidence for risk-outcome pairs, with higher star-ratings indicating stronger evidence of an association.

### Narrative synthesis

Narrative synthesis will be conducted by grouping studies according to exposure type and health outcome. We will explore the breadth of available evidence across groupings as well as highlight the health outcomes and violence types for which there is stronger evidence than others, drawing on results from meta-analyses and star-rating categories. The description of these patterns will allow us to make recommendations for future research as well as discuss the ways in which distinct types of violence affect health.

### Patient and public involvement

Patients and/or the public were not involved in the design, or conduct, or reporting, or dissemination plans of this research.

### ETHICS AND DISSEMINATION

The proposed review does not require formal ethical approval. Findings from this review will contribute to

GBD estimates of the health impact of GBV and VAC. The GBD includes data on morbidity and mortality from 1990 to present in 204 countries and territories for 369 diseases and injuries and 87 risk factors.[40] It is the most comprehensive worldwide observational epidemiological study to date and a critical tool used by clinicians, policymakers and researchers. Review findings will inform the GBD assessment of new risks and/or risk–outcome relationships and revisions to the magnitude of currently included associations. Updated global health estimates of the impact of GBV and VAC will be highlighted in consequent GBD releases and accompanying capstone publications.

In addition, this review is being conducted in conjunction with the Lancet Commission on GBV and the Maltreatment of Young People.[41] The aim of the commission is to complete a path breaking report on the global response to violence across the life-course, complete with new data insights and concrete policy and research recommendations that are informed by survivors and advocates. This report will be published by The Lancet in an effort to initiate debate, offer insight and explanation, and influence decision makers across the globe regarding GBV and VAC.

**Author affiliations**
[1]Institute for Health Metrics and Evaluation, University of Washington, Seattle, Washington, USA
[2]School of Nursing and Health Studies, University of Miami, Coral Gables, Florida, USA
[3]Warwick Medical School, University of Warwick, Coventry, UK
[4]University Libraries, University of Washington, Seattle, Washington, USA
[5]Barts and the London School of Medicine and Dentistry, Queen Mary University of London, London, UK
[6]Institute of Applied Health Research, University of Birmingham, Birmingham, UK

**Contributors** The initial PubMed search strategy was developed by CNS and refined and adapted to other databases by TJ. The writing and methodological plan for this protocol was developed by CNS, JSC, NM and EG. Further revisions to the protocol were made by all remaining authors (MJB, AD, MH, TJ, JKC, MK, RQHL and SM) and the final copy of the manuscript was approved by all authors.

**Funding** This work was supported by the Bill & Melinda Gates foundation grant number INV-018617.

**Disclaimer** The funders of had no role in the review design or the writing of the report.

**Competing interests** None declared.

**Patient and public involvement** Patients and/or the public were not involved in the design, or conduct, or reporting, or dissemination plans of this research.

**Patient consent for publication** Not applicable.

**Provenance and peer review** Not commissioned; externally peer reviewed.

**ORCID iDs**
Cory N Spencer http://orcid.org/0000-0001-6897-3157
Sonica Minhas http://orcid.org/0000-0001-9271-8623
Joht Singh Chandan http://orcid.org/0000-0002-9561-5141
Emmanuela Gakidou http://orcid.org/0000-0002-8992-591X
Nicholas Metheny http://orcid.org/0000-0003-2295-2945

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
