## [Reviewer comments · BMJ Open]

ARTICLE DETAILS

TITLE (PROVISIONAL)	Estimating the global health impact of gender-based violence and violence against children: a systematic review and meta-analysis protocol
AUTHORS	Spencer, Cory; Baeza, María Jose; Chandan, Jaidev Kaur; Debure, Alexandra; Herbert, Molly; Jewell, Teresa; Khalil, Mariam; Lim, Rachel Qian Hui; Minhas, Sonica; Chandan, Joht Singh; Gakidou, Emmanuela; Metheny, Nicholas

VERSION 1 – REVIEW

REVIEWER	Magalhães, Bruno Francisco Gentil Portuguese Institute for Oncology of Porto
REVIEW RETURNED	08-Mar-2022

GENERAL COMMENTS	Congratulations to the authors for their initiative and courage in proposing to develop a work of this dimension: for such a long period of inclusion of publications, for outcomes and the complexity. They comply with what is recommended by PRISMA-P for this type of publication. The intersection (AND) of the concepts “type of study” and measures of “risk” does not seem to make sense. Why did the authors operationalize the research in this way? I can’t understand why to search by type of study and measures of effect. How will the methodological quality of the included studies be evaluated?
--

REVIEWER	Hendrickson, Zoé Johns Hopkins University Bloomberg School of Public Health, Health, Behavior & Society
REVIEW RETURNED	15-Apr-2022

GENERAL COMMENTS	This is an exciting systematic review focus with great opportunity to impact understanding and estimates of experiences of violence globally. Please see my comments below on some suggestions or reflections for additional clarity that could be useful. Abstract: - More details on the methods in the abstract could be helpful – how many databases, steps (title/abstract review, full text, etc.) Background:
---

- It would be useful to expand the discussion of different forms of gender- and age-based violence in the first paragraph – and perhaps in a more central way in the background. There are specific terms/acronyms introduced throughout the introduction, and while many are consistently used, others define them differently. It would be helpful to understand exactly what the authors are understanding as part of age- and gender-based violence early on.
- Why has the GBD only focused on IPV and CSA to date? Further elaboration on why this has been restricted could help complete the current picture outlined in paragraph 2
- Can you elaborate on what you mean by “typically focused on targeted risk-outcome relationships” in p. 4, last paragraph? You elaborate on it later in the paragraph, but there could be a clearer way to explain what you mean here.
- P. 5, 2nd paragraph – would you include all child, early, and forced marriage as violence? Some define CEFM as part of gender-based violence.
- The third gap identified, “removing artificial barriers which have created isolated streams of research,” should be elaborated.
- More discussion of the lifecourse approach and its innovative approach would be helpful

Methods:

- Have the authors considered the language they are using related to victim vs. survivor vs. another term? Intentionality related to the language choice is recommended.
- There are a number of alternative terms related to FGM not included in the search strategy that could be considered
- Similar to my point above, child, early, and forced marriage (not just forced marriage) is understood by some as violence. Is this part of the search, or could it be?
- Do you have alternative terms used for each of these terms? For example, in addition to economic abuse, perhaps financial abuse or violence? Or, for example, psychological violence as well as psychological abuse?
- You discuss violence against children in the background, but it is not listed in the list of terms. Is there a reason for this?
- Is there a benefit to including specific acts (hitting, punching, etc.) as outlined by the WHO GBV tool in the list of terms?
- It makes sense that you are keeping the inclusion criteria open for “health outcomes,” but it may become problematic or confusing when reviewing titles/abstracts as people’s definitions of health may vary. Can you clarify how you will address this?
- Can you elaborate on the exclusion of cross-sectional studies and the justification? In addition, the reason for the exposure measurement exclusion criteria?
- Have the authors considered any sociological databases? SSCI for example?
- It says that search strategies were developed with controlled vocabulary, but it is clear that keywords or textwords were used as well?
- For the search strategy for risk, did you consider including “likelihood” or “likely” or similar?
- elaboration on the steps, for example that the titles/abstracts will be reviewed first, then full text review, should be clarified in the data management section.

	 - What if articles are retrieved that are not in the three languages listed (other than English)? - Why are all full-text articles not being reviewed by 2 authors? - Perhaps important to include details of how covidence will be used to document reasons for exclusion in the review process? - How will narrative synthesis be conducted? What methods will be used? How will data be synthesized in addition to the proposed potential meta-analysis and additional forest plots described?
--	--

VERSION 1 – AUTHOR RESPONSE

Reviewer: 1

Prof. Bruno Magalhães, Francisco Gentil Portuguese Institute for Oncology of Porto, Escola Superior de Saúde de Santa Maria

Comments to the Author:

Congratulations to the authors for their initiative and courage in proposing to develop a work of this dimension: for such a long period of inclusion of publications, for outcomes and the complexity. They comply with what is recommended by PRISMA-P for this type of publication.

The intersection (AND) of the concepts “type of study” and measures of “risk” does not seem to make sense. Why did the authors operationalize the research in this way? I can’t understand why to search by type of study and measures of effect.

Thank you for your question about the intersections of concepts within the search strings presented. We developed our search strategy by first identifying criteria by which we would screen studies and subsequently developing concept categories for each criteria. Our criteria included that studies (1) be longitudinal, case-control, or cross-over in nature; and (2) report an estimate of association. From these criteria, we developed the two concept categories, (1) ‘study type,’ and (2) ‘risk’, to guide our search. In testing our search strategy performance, we found that the inclusion of the ‘risk’ concept category in addition to study type refined our search results to epidemiological literature reporting estimates of association that we will ultimately be able to extract and analyze.

How will the methodological quality of the included studies be evaluated?

The methodological quality of the included studies will be evaluated through quality assessment during data extraction. We will rate each study according to six risk of bias criteria: exposure measurement, outcome measurement, representativeness of study population, control for confounding, selection bias, and reverse causation. Each criterion is detailed in the ‘Risk of bias in individual studies’ section of the manuscript.

Reviewer: 2

Dr. Zoé Hendrickson, Johns Hopkins University Bloomberg School of Public Health, Johns Hopkins University Center for Communication Programs

Comments to the Author:

This is an exciting systematic review focus with great opportunity to impact understanding and estimates of experiences of violence globally. Please see my comments below on some suggestions or reflections for additional clarity that could be useful.

Abstract:

-More details on the methods in the abstract could be helpful – how many databases, steps (title/abstract review, full text, etc.)

Thank you for your comment to include these additional details in the methods section of the abstract. In the revised abstract, we have listed by name the 7 databases included in the electronic search and further details surrounding the planned screening and data extraction processes of the review.

Manuscript changes:

“Methods and analysis: Electronic databases (PubMed, Embase, CINAHL, PsycInfo, Global Index Medicus, Cochrane, and Web of Science Core Collection) were searched from 1st January 1970 to 30th September 2021 and searches will be updated to the current date prior to final preparation of results. Reviewers will first screen titles and abstracts, and eligible articles will then be full-text screened and accepted should they meet all inclusion criteria. Data will be extracted using a standardized form with fields to capture study characteristics and estimates of association between violence exposure and health outcomes. Individual study quality will be assessed via six risk of bias criteria. For exposure-outcome pairs with sufficient data, evidence will be synthesized via a Meta-regression---Bayesian, regularized, trimmed model and confidence in the cumulative evidence assessed via the burden of proof risk function. Where possible, variations in estimates of association by subgroup, i.e. age, sex or gender, will be explored.”

Background:

-It would be useful to expand the discussion of different forms of gender- and age -based violence in the first paragraph – and perhaps in a more central way in the background. There are specific terms/acronyms introduced throughout the introduction, and while many are consistently used, others define them differently. It would be helpful to understand exactly what the authors are understanding as part of age- and gender-based violence early on.

Thank you for this feedback. We have updated the terms used throughout the manuscript to refrain from introducing new terminology that may cause confusion. In addition, we have included additional details within the first paragraph regarding the intersection of these fields.

Manuscript changes:

“Gender -based violence (including but not limited to intimate partner violence (IPV), elder abuse, and violence against women) and violence against children (VAC) are global public health issues associated with a substantial burden of morbidity and mortality. It is well known that the immediate consequences of both VAC and gendered-based violence in adulthood (GBV) can lead to physical injuries and death [1]. However, the medium- and longer-term consequences are less well understood, but have shown to span

a variety of physical, mental, sexual, and reproductive health issues [2,3]. Until recently, the fields of VAC and GBV were largely siloed, stunting our understanding of how different exposure to violence influence each other across the lifespan. To address these challenges, the Sexual Violence Research Initiative (SVRI), UNICEF Innocenti, and WHO have recently developed a framework of guiding principles encouraging interaction of research in the field of violence epidemiology [4]. As international advocacy and research organizations push for the integration of these fields, a more fulsome understanding of the health impacts of violence across the life-course is needed [5,6].”

-Why has the GBD only focused on IPV and CSA to date? Further elaboration on why this has been restricted could help complete the current picture outlined in paragraph 2

The GBD has so far been unable to expand the GBD estimates beyond IPV and CSA due to a lack of a systematic review such as the one proposed in this protocol. We attempt to make this clearer in paragraph 2 and further make the case for the need to complete this review.

Manuscript changes:

“While these findings provide a basis for understanding the impact of violence on health, the lack of a comprehensive analysis of the longitudinal literature has so far precluded the ability to expand the types of violence included in the GBD as well as the specific health outcomes that comprise the final estimates of burden. A more complete understanding of the adverse health outcomes associated with exposure to more types of GBV and VAC, and the magnitude of these associations, is needed to capture the negative health and societal impacts of GBV and VAC.”

-Can you elaborate on what you mean by “typically focused on targeted risk-outcome relationships” in p. 4, last paragraph? You elaborate on it later in the paragraph, but there could be a clearer way to explain what you mean here.

We have removed this phrasing and instead use clearer language to illustrate our point that existing reviews usually look at the impact of one type of violence on a single health outcome.

Manuscript changes:

“Beyond the estimates provided by the internationally comparative GBD studies, existing reviews assessing the health impacts of GBV and VAC have typically focused on the impact of a single type of violence (e.g. IPV) on a specific health outcome (e.g. HIV).”

-P. 5, 2nd paragraph – would you include all child, early, and forced marriage as violence? Some define CEFM as part of gender-based violence.

We also do include CEFM as a part of gender-based violence. However, we appreciate that this was previously not clearly captured. As such, we have now added additional terms to our search strategy to

capture the multiple ways CEFM is referenced in the literature. We have updated the example definitions of violence provided in the manuscript to reflect these changes as well.

Manuscript changes:

"We include in our searches the following terms describing exposure to GBV and/or VAC:

- *Gender-based violence;*
- *Intimate partner violence, partner abuse/violence, wife/spouse abuse, dating abuse/violence;*

- *Sexual abuse, rape, forced sex, sexual assault, sexual coercion, sexual exploitation;*
- *Reproductive coercion;*
- *Female genital mutilation, female genital cutting, female circumcision;*
- *Sex trafficking, child, early and forced marriage;*
- *Physical abuse;*
- *Psychological abuse, emotional abuse, verbal abuse;*
- *Economic abuse, financial abuse;*
- *Cyberviolence, cybervictimization;*
- *Domestic violence/abuse;*
- *Adverse childhood experiences that include direct exposure to and witnessing of violence;*
- *Child maltreatment, molestation, child abuse;*
- *Elder abuse, senior abuse, aged abuse;*
- *Stalking, cyberstalking;*
- *Dehumanization, torture;*
- *Workplace violence, student abuse, sexual harassment;*
- *Gender-based violence perpetrated with a firearm.”*

-The third gap identified, “removing artificial barriers which have created isolated streams of research,” should be elaborated.

We now rephrase this to be more concise and precise.

Manuscript changes:

“These include the quantification of the health burden of less-studied forms of violence, the health burden of violence in in lower- and middle-income settings, as well as the integration of violence in childhood and adulthood as an intergenerational issue that could be more effectively measured using a life-course approach.”

-More discussion of the lifecourse approach and its innovative approach would be helpful

We agree and have updated the background to include a more explicit description of this approach and its benefits within the field of violence epidemiology.

Manuscript changes:

“These include the quantification of the health burden of less-studied forms of violence, the health burden of violence in in lower- and middle-income settings, as well as the integration of violence in childhood and adulthood as an intergenerational issue that could be more effectively measured using a life-course approach. The life-course approach as outlined in the Minsk Declaration which essentially recognizes that all stages of a person's life are intricately intertwined with each other, with the lives of other people in society, and with past and future generations of their families [22,23]. In order to do adopt this approach effectively when considering the health effects of GBV/VAC, we must consider that violence can occur at any stage in one's life (pre-conception to death) but also the impact of such event can be inter-generational and societal. Additionally, as highlighted through the reviews cited above, the current research trajectory often creates distinctions between GBV/VAC and other forms of life course violence such as elder-abuse [18]. However, considering that GBV and VAC share risk factors, co-occur and can lead to compounding consequences across the life course, there is a clear need to examine these phenomena in unison.”

Methods:

-Have the authors considered the language they are using related to victim vs. survivor vs. another term? Intentionality related to the language choice is recommended.

As violence researchers, we wrestle with the right term to use in this situation and understand the different implications each can have. We choose to use victim throughout this protocol for several reasons. First, the term "victimization" is included throughout our search terms and using "victim" allows for parsimony in our writing. Second, it made sense to use "victim" when researching the health consequences of violence, given the negative impact violence has on most of the health outcomes studied. Were we studying resilience factors or otherwise promoting a strengths-based approach, survivor may be more appropriate. Third, "victim" is the language used most often throughout the literature we are searching. Given the epidemiological nature of this study, it made sense to keep the language consistent. Finally, and perhaps most disturbingly, we include studies of gender-based homicide, suicide, and other mortality outcomes. This means that not all those included in this review are "survivors."

-There are a number of alternative terms related to FGM not included in the search strategy that could be considered

Thank you for this suggestion. Using PubMed as an example database, we tested including the search terms "infibulation*," "clitoridectom*," "clitorectom*," "ritual female genital surger*" and "FGM" to the exposure concept of our search strategy. All other aspects of the search string kept the same, the inclusion of these additional terms yielded 90 more articles. We have updated the documentation of our search strategies to reflect the newly included terms.

-Similar to my point above, child, early, and forced marriage (not just forced marriage) is understood by some as violence. Is this part of the search, or could it be?

Thank you for your suggestion. We also do include CEFM as a part of gender-based violence. However, we appreciate that this was previously not fully captured by our search terms. Using PubMed as an example database, we tested including the search terms "early marriage*," "child marriage*," "child bride*," and "CEFM" to the exposure concept of our search strategy. All other aspects of the search string kept the same, the inclusion of these additional terms yielded 105 more articles. We have updated the documentation of our search strategies to reflect the newly included terms.

-Do you have alternative terms used for each of these terms? For example, in addition to economic abuse, perhaps financial abuse or violence? Or, for example, psychological violence as well as psychological abuse?

Thank you for these suggestions. We have included 'violence' as a controlled vocabulary and title/abstract keyword term in our search strategy, which will encompass literature referencing 'financial violence' or 'psychological violence' specifically. We chose to use 'violence' as a search term without additional modifiers in order to capture the myriad different ways violence against children, young people, women, and other populations can be referenced.

We have additionally tested adding the suggested term, “financial abuse”, to our searches. Using PubMed as an example, we found that this addition yielded 2 more articles, with all other aspects of the search string the same. We have updated the documentation of our search strategies to reflect our addition of this term.

-You discuss violence against children in the background, but it is not listed in the list of terms. Is there a reason for this?

Thank you for this question. We have included ‘violence’ as a controlled vocabulary and title/abstract keyword term in our search strategy, which will encompass literature referencing ‘violence against children’ specifically. We chose to use ‘violence’ as a search term without additional modifiers in order to capture the myriad different ways violence against children, young people, women, and other populations can be referenced.

-Is there a benefit to including specific acts (hitting, punching, etc.) as outlined by the WHO GBV tool in the list of terms?

We appreciate the Reviewer’s question on this topic and we agree with the importance of using acts-based questions in order to assess exposure to violence in survey tools. Many gold-standard GBV tools use acts-based questions to measure categories of exposure (i.e. physical, sexual) to violence. In epidemiological studies, individual acts/survey questions from GBV tools are typically operationalized into an exposure category which is characterized by type of violence and perpetrator identity rather than by specific act. In some cases, specific acts are reported in addition to operationalized categories. By including terms for these exposure categories, our search strategy should identify literature that report both exposure categories and specific acts. For literature that only reports specific acts, while we have not included these terms explicitly in our search, it is likely that these literature are catalogued with broader controlled vocabulary terms or reference general keywords such as ‘violence’ in the title/abstract.

-It makes sense that you are keeping the inclusion criteria open for “health outcomes,” but it may become problematic or confusing when reviewing titles/abstracts as people’s definitions of health may vary. Can you clarify how you will address this?

Thank you for highlighting this important point, and we agree with the Reviewer’s perspective that this characteristic of our search can be a source of confusion in title/abstract screening. In title/abstract screening, reviewers reference the Global Burden of Diseases study definitions of risk factors, injuries, and diseases (<https://www.thelancet.com/gbd/summaries>) to evaluate whether a study definition is an eligible health outcome. In addition, all reviewers meet regularly to discuss and clarify points of confusion, including interpretations of eligible health outcomes. Group decisions made during these meetings are documented and circulated so that reviewer decisions remain consistent with regard to definitions of health outcomes. We have included additional detail describing this process in the revised manuscript.

Manuscript changes:

“We did not restrict searches to pre-defined health outcomes and aim to accept all literature reporting an association between violence exposure and health. Definitions of health outcomes and health-related risk factors will be guided by causes, injuries, and risk factor case definitions from the Global Burden of Diseases study [8,25]. Studies that report on certain biomarkers or without accompanying clinical

diagnoses (i.e., neural connectivity patterns, salivary cortisol as a stress response, DNA characteristics) will not be eligible for inclusion. Similarly, studies that report on the presence or number of disease symptoms without an accompany diagnosis of a health outcome will not be eligible for inclusion. Reviewers will meet regularly to raise questions about eligible health outcomes, with consensus decisions documented and circulated via written guidelines. Differences in measurement methods or case definitions of eligible health outcomes will be documented as a part of quality assessment as well. Final selection of associations to be synthesized will depend on the availability of studies that examine the association between a comparable form of exposure and reported health outcome.”

-Can you elaborate on the exclusion of cross-sectional studies and the justification? In addition, the reason for the exposure measurement exclusion criteria?

Thank you for your questions. We exclude cross-sectional studies in accordance with Global Burden of Disease study risk factor analyses, which typically do not include cross-sectional studies. This exclusion reason is related to the inability to assess temporality between exposures and outcomes in cross-sectional studies. Additionally, we exclude studies for which only aggregate measure of exposure which combine a form of violence exposure with other, non-eligible exposures due to the specific nature of our review and analysis goals. For example, many studies report adverse childhood experience (ACE) exposure as an aggregate of many types of ACEs, including household challenges that are not direct exposures to violence and outside of the scope of this review. For these studies, it is not possible to disentangle the effects of violence from the effects of other hardships and we are therefore unable to include them in our review.

We have included additional clarity on these aspects of our exclusion criteria in the Exclusion section of the main text.

Manuscript changes:

“Exclusion

- *Study design: Cross-sectional, ecological, case series or case studies. We exclude cross-sectional studies in accordance with Global Burden of Disease study risk factor analyses, which typically do not include cross-sectional studies. This exclusion reason is related to the inability to assess temporality between exposures and outcomes in cross-sectional studies. We do not anticipate there to be any experimental studies, however, these will also be excluded.*
- *Participants: Studies conducted in subgroups identified only by convenience sampling or subgroups identified via a shared characteristic that is likely related to risk of exposure to violence or the reported health outcome (e.g. domestic violence shelter residents).*
- *Exposure measurement: Studies that report only an aggregate measure of exposure combining exposure to a form of violence with other, non-eligible exposures (e.g., reports a composite adverse childhood experience score only) will be excluded. For these studies, we are unable to disentangle the effect of violence exposure from the effects of other hardships or exposure types, preventing their inclusion in our review.*

- *Does not meet minimum reporting criteria: Studies missing essential data, i.e., do not report effect sizes and uncertainty information (confidence intervals, sample sizes) or the data needed to impute an effect size with uncertainty information.*
- *Studies reporting on the same exposure and outcome using the same data: The study with the longest follow-up time period or most complete dataset will be included.”*

-Have the authors considered any sociological databases? SSCI for example?

Thank you for your suggestion. We searched the Web of Science Core Collection, which includes SSCI. We have revised our mention of Web of Science in the manuscript to “Web of Science Core Collection” to better indicate the databases captured by our search.

-It says that search strategies were developed with controlled vocabulary, but it is clear that keywords or textwords were used as well?

Thank you for raising this question. It is correct that we included both controlled vocabulary and keyword search terms in our search strategies, and we have revised our description of the search process to reflect this aspect of our search.

Manuscript changes:

“PubMed, Embase, CINAHL, PsycInfo, Global Index Medicus, Cochrane, and Web of Science Core Collection were searched using controlled vocabulary and keyword search terms.”

-For the search strategy for risk, did you consider including “likelihood” or “likely” or similar?

Thank you for your suggestion. Using PubMed as an example database, we tested including the search terms “likel*” to the risk concept of our search strategy. All other aspects of the search string kept the same, the inclusion of these additional terms yielded 592 more articles. We have updated the documentation of our search strategies to reflect the newly included terms.

-elaboration on the steps, for example that the titles/abstracts will be reviewed first, then full text review, should be clarified in the data management section.

Thank you for your suggestion. We have made several adjustments to the data management and extraction section to clarify the order of the review steps.

Manuscript changes:

“Reviewers will complete title and abstract screening of all articles before the team proceeds to full-text screening. Studies that met inclusion criteria in title and abstract screening will additionally be full-text screened and excluded if found to meet any of the exclusion criteria. Following the PRISMA guideline, each exclusion will include documentation of a specific exclusion reason. Within Covidence, there are several built-in exclusion reasons (i.e., wrong study design, wrong setting, etc.) as well as the possibility to create custom exclusion reasons. Reviewers will meet to discuss the addition of custom exclusion reasons prior to beginning full-text review, and will iteratively meet to discuss addition of new reasons as necessary. For full text review, 10% of articles will be reviewed by two independent reviewers and a

meeting held to resolve misunderstandings and ensure all reviewers clearly understand inclusion and exclusion criteria. The remaining 90% of articles will be reviewed by one independent reviewer. If reviewers are unable to access the full text of a publication, the reviewers will reach out directly to the corresponding author and wait a maximum of one month for response, after which point the article will be excluded.

Data extraction will occur in parallel with full-text review, with some team members beginning to extract data once a sufficient number of full-text articles have been accepted. Before any reviewer begins data extraction, the entire review team will conduct a consensus building exercise in which all reviewers extract data from the same 10 accepted articles. In a group meeting, extractions will be compared and

any questions resolved. Reviewers will extract data from accepted articles using a standardized form created in Covidence [26].”

-What if articles are retrieved that are not in the three languages listed (other than English)?

Thank you for raising this point. For such articles, we plan to reach out to colleagues fluent in languages other than English, Spanish, French, and Portuguese to assist with full-text screening and extraction. We have added this intention within the data management and extraction section of the manuscript.

Manuscript changes:

“Non-English publications will be reviewed using the language fluencies (Spanish, French, and Portuguese) of the reviewers. Should articles in other languages be retrieved and eligible for extraction, the reviewers will contact colleagues fluent in these languages for assistance.”

-Why are all full-text articles not being reviewed by 2 authors?

Thank you for this question. Due to the breadth of our search, we expect a large number of articles to be accepted for full-text screening. With a limited team of reviewers available to screen each article, we sought to balance the priorities of independent review and completing our review in a timely manner. Thus, we have chosen an approach to screening that aims to build consensus among reviewers and achieve low conflict rates at the beginning of each stage before proceeding with single author screening. In the title and abstract screening stage, the first two-thirds of articles were double-screened and we observed a low conflict rate of 6%. We therefore proceeded with single screening for the remaining one third of articles. Similarly, full-text screening will begin by having 2 authors review the first 10% of eligible articles. After this stage, we will proceed to having one author review the remaining 90% of articles, conditional on the observation of a low conflict rate (<10%). During initial double-screening periods, we meet regularly to discuss points of confusion or questions, promoting consistent screening decisions among reviewers that will carry on to the remainder of single-screened articles.

-Perhaps important to include details of how covidence will be used to document reasons for exclusion in the review process?

Thank you for your suggestion. We have revised the data management and extraction section to include details on how reasons for exclusion in full-text screening will be documented in Covidence.

Manuscript changes:

“Reviewers will complete title and abstract screening of all articles before the team proceeds to full-text screening. Studies that met inclusion criteria in title and abstract screening will additionally be full-text screened and excluded if found to meet any of the exclusion criteria. Following the PRISMA guideline,

each exclusion will include documentation of a specific exclusion reason. Within Covidence, there are several built-in exclusion reasons (i.e., wrong study design, wrong setting, etc.) as well as the possibility to create custom exclusion reasons. Reviewers will meet to discuss the addition of custom exclusion reasons prior to beginning full-text review, and will iteratively meet to discuss addition of new reasons as necessary.”

-How will narrative synthesis be conducted? What methods will be used? How will data be synthesized in addition to the proposed potential meta-analysis and additional forest plots described?

Thank you for your questions. Our narrative synthesis will be conducted by first grouping studies by health outcome measured and then by type of violence measured. This will allow us to better understand the breadth of evidence and ascertain whether there are certain health outcomes and types of violence that have stronger evidence than others, allowing us to highlight recommendations for future research as well as how distinct types of violence affect health and how violence (write large) impacts discrete health outcomes. This narrative synthesis will be distilled graphically using harvest plots, which allow for the summary of heterogeneous evidence that shows how violence is associated with a particular health outcome (or class of outcomes), as well as patterns by study quality and evidence gaps. We will use elements of the Systematic Review without Meta-Analysis (SWiM) guidelines to report these findings. This approach calls for the documentation of narrative synthesis methods across nine criteria, including how studies are grouped, the standardized metric used for the synthesis, the synthesis method, and how data are presented. Though designed to improve reporting and transparency in reviews that do not employ meta-analyses, they will ensure we report all relevant findings systematically.

We have added an additional section to our data synthesis plan to explicitly describe these methods in the manuscript.

Manuscript changes:

“Narrative Synthesis

Narrative synthesis will be conducted by grouping studies according to exposure type and health outcomes. We will explore the breadth of available evidence across groupings as well as highlight the health outcomes and violence types for which there is stronger evidence than others, drawing upon results from meta-analyses and risk-outcome scores. The description of these patterns will allow us to make recommendations for future research as well as discuss the ways in which distinct types of violence affect health.”

VERSION 2 – REVIEW

REVIEWER	Hendrickson, Zoé Johns Hopkins University Bloomberg School of Public Health, Health, Behavior & Society
REVIEW RETURNED	01-Jun-2022
GENERAL COMMENTS	I have carefully reviewed the revised submission and do not have additional comments to make at this juncture. The protocol is ready to be accepted based upon my review. I am grateful to the authors for having considered my many comments and for having responded so thoughtfully to them all.